# Factors of parental COVID-19 vaccine hesitancy: A cross sectional study in Japan

Sayaka Horiuchi[1]*, Haruka Sakamoto[2], Sarah K. Abe[3], Ryoji Shinohara[1], Megumi Kushima[1], Sanae Otawa[1], Hideki Yui[4], Yuka Akiyama[4], Tadao Ooka[4], Reiji Kojima[4], Hiroshi Yokomichi[4], Kunio Miyake[4], Takashi Mizutani[5], Zentaro Yamagata[4]

1 Center for Birth Cohort Studies, University of Yamanashi, Chuo-shi, Yamanashi, Japan, 2 Department of Global Health Policy, University of Tokyo, Bunkyo-ku, Tokyo, Japan, 3 Division of Prevention, National Cancer Center Institute for Cancer Control, Tokyo, Japan, 4 Department of Health Sciences, School of Medicine, University of Yamanashi, Chuo-shi, Yamanashi, Japan, 5 Minami Nagata Clinic, Yokohama, Japan

* sayakahoriuchi@gmail.com

**Data Availability Statement:** All relevant data are within the paper and its Supporting Information files.

## Abstract

The eligibility of COVID-19 vaccines has been expanded to children aged 12 and above in several countries including Japan, and there is a plan to further lower the age. This study aimed to assess factors related to parental COVID-19 vaccine hesitancy. A nationwide internet-based cross-sectional study was conducted between May 25 and June 3, 2021 in Japan. The target population was parents of children aged 3–14 years who resided in Japan, and agreed to answer the online questionnaire. Parental COVID-19 vaccine hesitancy (their intention to vaccinate their child) and related factors were analyzed using logistic regression models. Interaction effects of gender of parents and their level of social relationship satisfaction related to parental vaccine hesitancy was tested using log likelihood ratio test (LRT). Social media as the most trusted information source increased parental vaccine hesitancy compared to those who trusted official information (Adjusted Odds Ratio: aOR 2.80, 95% CI 1.53–5.12). Being a mother and low perceived risk of infection also increased parental vaccine hesitancy compared to father (aOR 2.43, 95% CI 1.57–3.74) and those with higher perceived risk of infection (aOR 1.55, 95% CI 1.04–2.32) respectively. People with lower satisfaction to social relationships tended to be more hesitant to vaccinate their child among mothers in contrast to fathers who showed constant intention to vaccinate their child regardless of the level of satisfaction to social relationship (LRT p = 0.021). Our findings suggest that dissemination of targeted information about COVID-19 vaccine by considering means of communication, gender and people who are isolated during measures of social distancing may help to increase parental vaccine acceptance.

## Introduction

The coronavirus disease 2019 (COVID-19) pandemic has led to increasing cases and death tolls, and strained health care systems across the globe [1]. Regardless of measures to prevent transmission such as hand-washing, mask-wearing, travel restrictions and social distancing,

**Funding:** The authors received no specific funding for this work.

**Competing interests:** The authors have declared that no competing interests exist.

many countries continue to experience surges of cases. In Japan, as of October 25, 2021, the total number of cases and deaths has reached 1,716,938 and 18,199 respectively [2]. Along with the measures to prevent transmission, a COVID-19 vaccination is expected to save countless lives, reduce the burden on healthcare systems and mitigating the outbreak.

COVID-19 vaccines were rapidly developed, and Pfizer-BioNTech COVID-19 vaccine and the Moderna COVID-19 vaccine were approved in Japan on February 14, 2021 and on May 21, 2021 [3]. The vaccine program in Japan launched in February first targeting medical care workers and people aged 65 years and above, and then rolled out to all under 65 in May. In June 2021 the Minister of Health, Labor and Welfare expanded eligibility to youth aged 12 and above [4]. As of October 18, 2021, 40.4% of children aged 12 to 19 years completed a second dose of the vaccine while 90.1% of people aged 65 and above [5]. A proportion of new infections occurred in people aged 10 to 30 accounts for more than half of the total population [6]. Therefore, reaching those who remain vaccine hesitant among the younger generation has become increasingly important in order to control the outbreak. The young generation with children who are eligible to receive the COVID-19 vaccine would have significant influence on their child' vaccine uptake. To increase the vaccination rate among minors in Japan, parental hesitancy must be considered.

Globally, reasons for vaccine hesitancy among the general public are categorized into (1) the risk-benefit of vaccines, (2) knowledge and awareness issues, (3) religious, cultural, gender or socio-economic factors, and fear of adverse reaction were cited as major issue [7]. Several studies examined vaccine hesitancy among adults in Japan reported that female, younger age (compared to those aged 65 and above), those with lower socio-economic status and those without perceived risk of infection were more likely to report vaccine hesitancy [8–11]. Along with expansion of the COVID-19 vaccine program to children, it has become important to understand factors associated with parental vaccine hesitancy and COVID-19 vaccines whether there are factors unique to COVID-19 vaccines. The present study, therefore, aimed to assess the proportion and characteristics of parents who reported hesitancy towards COVID-19 vaccines.

## Materials and methods

This study was conducted according to the principles of the World Medical Association Declaration of Helsinki and Ethical Guidelines for Medical and Health Research Involving Human Subjects promulgated by the Ministry of Health, Labor, and Welfare of Japan. This study was approved by the Ethics Review Board of the University of Yamanashi (approval number: 2259). Online informed consent was obtained from all participants who met eligibility criteria after filling in screening questions. Participants clicked the checkbox on the screen online to provide informed consent.

### Design and setting

The University of Yamanashi performed a nationwide internet-based cross-sectional study between May 25 and June 3, 2021. The current study is part of a follow-up study that was implemented to evaluate psychological status of caregivers and children in 2020. This study was done during the state of emergency and after the vaccine became available to the general population. Under a "state of emergency", the government requests people and private entities to restrict travel, commuting, social gatherings, and use of facilities without any legally binding obligations.

### Participants

The eligibility criteria were 1) those who had children aged 3–14 years, and 2) those who resided in Japan at the time of inception of the survey. We did not set an age limit for

participants. We targeted children under 14 because this group will be more influenced by parental decision compared to older children. Also, COVID-19 vaccine may be expanded to children aged 5 and more in coming years. Participants were recruited from those who had voluntarily registered with a panel of the Nippon Research Center for web-based surveys in response to online affiliate advertising [12]. The Nippon Research Center is a private entity that subcontracts public and private surveys. To respond needs of its customers, it has built several panels and recruited participants from whole nation. As of October 2020, about 1.4 million registered in the panel. Among them, 56% were female, and median age was in 40 [13]. Those who answered screening questions and met the eligibility criteria were asked to participate in the survey. Only those who provided their consent on the website answered the entire questionnaire.

## Data collection

The internet-based questionnaire consisted of 43 questions including screening questions related to parents and their children. It was developed under supervision of public health professionals. Data collection was commissioned by the Nippon Research Center. People registered in the panel participated in this survey on a voluntary basis, in response to an invitation. Data collection continued until the sample size reached 1,200. The sample size was considered sufficient to detect a 10% absolute difference in the percentage of participant's intention to vaccinate their child across different trusted source of news media based on reports of a 20–30% prevalence of vaccine hesitancy in the general population during the pandemic [10,11,14].

If respondents had more than one child within the target age, they were asked to restrict their answers to one child, who was randomly selected by the system.

## Outcome

The primary outcome was parents' intention to vaccinate their child. The parent answered "Yes" or "No" to the question asking whether they would like their child to receive a COVID-19 vaccination when it became available.

## Potential factors related to vaccine hesitancy

Factors related to vaccine hesitancy have been reported as demographic factors, socio-economic status, perceived risks and concerns over vaccine safety and effectiveness, perceived risks of COVID-19, sources of information about COVID-19 [14–16]. Parental intention to vaccine themselves was also reported to be a strong factor of willingness to vaccine their children. Accordingly, we selected potential factors for vaccine hesitancy as follows.

Parents' demographic factors included parents' gender and age, child's gender and age, size of residential city, state of emergency declared in the residential area at the time of survey. Socioeconomic factors included job type, partner's job type, impact of COVID-19 on their work, household income in 2020. We categorized the job type into employed/self-employed, part-time, and unemployed/homemaker/student. Impact of COVID-19 on work was measured as whether a person was fired or had their salary reduced after the outbreak.

Several psychological characteristics were captured. Trusted source of information was asked to choose the most trusted source for information on COVID-19 (Please choose one source of information you trust the most.). Satisfaction regarding social relationships including family, friends and coworkers was measured on a scale ranging from 0 (not satisfied at all) to 10 (very satisfied). Perceived risk of infection was measured with five questions regarding their concerns about getting infected. The perceived risk of infection was recorded high if a parent selected three and more, medium for one or two and low for zero. Distrust of the health

care system was measured with three questions regarding their concerns about health system amid of outbreak. Distrust of the health care system was recorded high if a parent selected two and more, medium for one and low for zero. Mental health status was assessed using the Japanese version of the Kessler Psychological Distress Scale (K6) [17,18]. The K6 has demonstrated excellent internal consistency and reliability and is widely used in epidemiological studies [19,20].

### Data analysis

The distribution of each variable was analyzed and summarized as number (%) (Table 1). We assessed the association between parent's intention to vaccinate themselves and intention to vaccinate their child using chi-squared test. We descriptively summarized reasons why parents did not intend to vaccinate themselves. We performed univariate and multivariate analysis to analyze the association between intention to vaccinate child and each of the covariates using logistic regression models. We selected variables to be included in the multivariate model based on a-priori hypothesis, and included all covariates in the model unless there was an evidence of multicollinearity. Finally, we tested interaction in effects of gender of parents and their satisfaction regarding social relationships during the outbreak on parent's intention to vaccinate their child using log likelihood ratio test. Stratum specific odds ratios (stratified by gender of parents and satisfaction regarding social relationship) and their 95% confidence intervals were estimated. STATA/MP 16.1 software was used for all analyses.

### Results

We requested a total of 2648 people to answer the questionnaire. Among 1745 people who answered screening questions (response rate: 65.9%), we excluded 545 people (Fig 1) because they did not meet eligibility criteria. There were no missing data.

Among 1,200 parents, 424 (35.3%) did not intend to vaccinate their child and 315 (26.3%) themselves when the vaccine became available (Table 1). About half of the parents were women (49.1%). The most trusted source of information regarding COVID-19 was government/public organization (26.7%) followed by private news media (22.3%). About one quarter of parents (24.4%) and 16.3% of their partners experienced loss of job or salary reduction. One fifth (19.2%) had low level of perceived risk of infection.

There was a strong relationship between intention to vaccinate themselves and their children (chi-squared test: p<0.001), and only 10.8% of those who did not want to vaccinate themselves had intention to vaccinate their children while 83.8% of those who wanted to vaccinate themselves did (Table 2). The main reason of having no intention to vaccinate themselves was fear of adverse reaction and safety of COVID-19 vaccines (201/315) (Fig 2).

The univariate analyses showed that trust in sources of COVID-19 related information other than government/public organization or public news media, female gender either of parent or child, younger age of parent (<34 years), lower household income, unemployment, lower perceived risk of infection and younger age of children were associated with higher odds of parental vaccine hesitancy (Table 3). After adjusting for covariates, social media as a trusted source of information, being a mother, lower perceived risk of infection and younger age of children were still associated with higher odds of parental vaccine hesitancy. Compared to those who trusted official information, those who trusted social media as a source of information were three times more likely to show no intention to vaccinate their child (OR 2.80, 95% CI 1.53–5.12). Mothers and people with low perceived risk of infection were twice the time more likely to report vaccine hesitancy compared to fathers (OR 2.43, 95% CI 1.57–3.74) and those who had high perceived risk of infection (OR 1.55, 95% CI 1.04–2.32) respectively.

**Table 1. Characteristics of study parents in Japan (year 2021) ($N$ = 1,200).**

| Variable | Number (%) |
|---|---|
| **Intention to vaccine their child** | |
| Yes | 776 (64.7) |
| No | 424 (35.3) |
| **Intention to vaccinate themselves** | |
| Yes | 885 (73.8) |
| No | 315 (26.3) |
| **Trusted source of information** | |
| Government/public organization | 320 (26.7) |
| Public news media (NHK) | 200 (16.7) |
| Private news media | 267 (22.3) |
| Social media | 60 (5.0) |
| Word of mouths | 101 (8.4) |
| No source | 252 (21.0) |
| **Parent's gender** | |
| Male | 611 (50.9) |
| Female | 589 (49.1) |
| **Parent's age (years)** | |
| < 34 | 111 (9.3) |
| 35–39 | 271 (22.6) |
| 40–45 | 333 (27.8) |
| ≥ 45 | 485 (40.4) |
| **Annual household income in 2020 (million yen)** | |
| <4 | 166 (13.8) |
| 4–6 | 309 (25.8) |
| ≧6 | 656 (54.7) |
| Unknown | 69 (5.8) |
| **Parent's job type** | |
| Employed/self-employed | 750 (62.5) |
| Part-time | 163 (13.6) |
| Unemployed/homemaker/student | 287 (23.9) |
| **Partner's job type** | |
| Employed/self-employed | 747 (62.3) |
| Part-time | 211 (17.6) |
| Unemployed/homemaker/student | 196 (16.3) |
| No partner | 46 (3.8) |
| **Parent's work place** | |
| Work at home everyday | 125 (10.4) |
| Work at home half of a week | 115 (9.6) |
| Work away from home | 670 (55.8) |
| Unemployed/homemaker/student | 290 (24.2) |
| **Partner's work place** | |
| Work at home everyday | 97 (8.1) |
| Work at home half of a week | 95 (7.9) |
| Work away from home | 759 (63.3) |
| Unemployed/homemaker/student | 203 (16.9) |
| No partner | 46 (3.8) |
| **Fired or salary reduction after the outbreak (parent)** | |

(*Continued*)

**Table 1.** (Continued)

| Variable | Number (%) |
|---|---|
| No | 907 (75.6) |
| Yes | 293 (24.4) |
| **Fired or salary reduction after the outbreak (partner)** | |
| No | 959 (79.9) |
| Yes | 195 (16.3) |
| No partner | 46 (3.8) |
| **Parent's satisfaction to the social relationship** | |
| High | 221 (18.4) |
| Medium | 675 (56.3) |
| Low | 304 (25.3) |
| **Perceived risk of infection** | |
| High | 512 (42.7) |
| Medium | 458 (38.2) |
| Low | 230 (19.2) |
| **Distrust of the health care system** | |
| Low | 738 (61.5) |
| Medium | 216 (18.0) |
| High | 246 (20.5) |
| **Parent's mental distress (K6 score)** | |
| None (0–4) | 633 (52.8) |
| Moderate (5–9) | 250 (20.8) |
| Severe ($\geq$ 10) | 317 (26.4) |
| **Concerns about household economic situation** | |
| High | 177 (14.8) |
| Medium | 311 (25.9) |
| Low | 712 (59.3) |
| **Concerns about own and family member's stress** | |
| High | 176 (14.7) |
| Medium | 288 (24.0) |
| Low | 736 (61.3) |
| **Concerns about lifestyle of their child** | |
| High | 151 (12.6) |
| Medium | 204 (17.0) |
| Low | 845 (70.4) |
| **Size of city of residence (population)** | |
| < 50,000 | 149 (12.4) |
| 50,000–100,000 | 151 (12.6) |
| 100,000–300,000 | 288 (24.0) |
| 300,000–500,000 | 155 (12.9) |
| $\geq$ 500,000 | 457 (38.1) |
| **State/Semi-state of emergency is declared at the time of survey** | |
| Yes | 872 (72.7) |
| No | 328 (27.3) |
| **Living with respondent's/partner's parents** | |
| No | 2,300 (95.8) |
| Yes | 100 (4.2) |
| **Number of children living together** | |

(*Continued*)

**Table 1.** (Continued)

| Variable | Number (%) |
|---|---|
| 1 | 390 (32.5) |
| 2 | 621 (51.8) |
| $\geq$ 3 | 189 (15.8) |
| **Child's gender** | |
| Male | 624 (52.0) |
| Female | 576 (48.0) |
| **Child's age (years)** | |
| 3–5 | 400 (33.3) |
| 6–11 | 500 (41.7) |
| $\geq$ 12 | 300 (25.0) |

The effect of parent's satisfaction regarding social relationship on intention to vaccinate their child varied by parent's gender (LRT: $p = 0.021$) (Table 4). While fathers had constant or more intention to vaccinate their child at the lower level of satisfaction, mothers were more likely to hesitate to vaccinate their child when they had lower satisfaction regarding social relationship (Fig 3).

## Discussion

The present study showed that high agreement between parent's intention to vaccinate themselves and their child, which is in line with a previous study [21]. On the other hand, about 16.2% of parents who wanted to get vaccinated themselves did not intend to vaccinate their child. This indicates that promoting parent's acceptance by providing information may lead to wider coverage of vaccination for children.

The present study showed that trust in resources other than public information, being a mother and low perceived risk of infection were associated with parental hesitancy to vaccinate

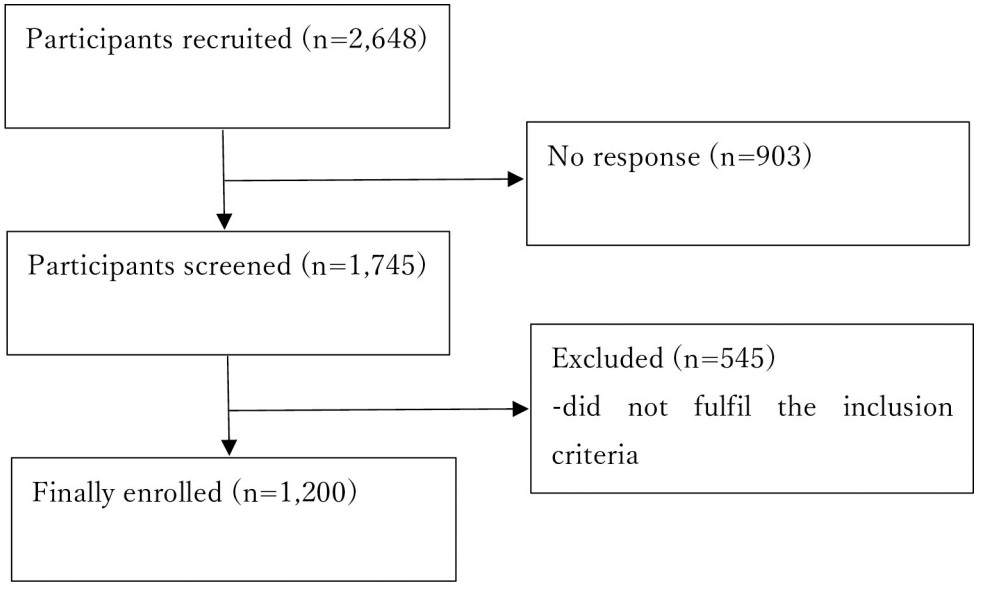

**Fig 1. Participant flow.**

**Table 2. Association between parents' intention to vaccinate themselves and intention to vaccinate their child in Japan (year 2021) (N = 1,200).**

|  | Intention to vaccinate their child | | p-value (Chi-square test) |
| --- | --- | --- | --- |
|  | **Yes** | **No** |  |
| **Intention to vaccinate themselves** |  |  |  |
| Yes | 742 (83.8) | 143 (16.2) | <0.001 |
| No | 34 (10.8) | 281 (89.2) |  |
| **Total** | 776 (64.7) | 424 (35.3) |  |

The numbers are number (%) unless otherwise indicated.

their child. The results are in line with previous studies that examined vaccine hesitancy among the general population and parents. Several studies reported that distrust in public information and use of social media were associated with vaccine hesitancy [14,22,23]. Misinformation is shared and rapidly circulated on social media. People supporting anti-vaccination

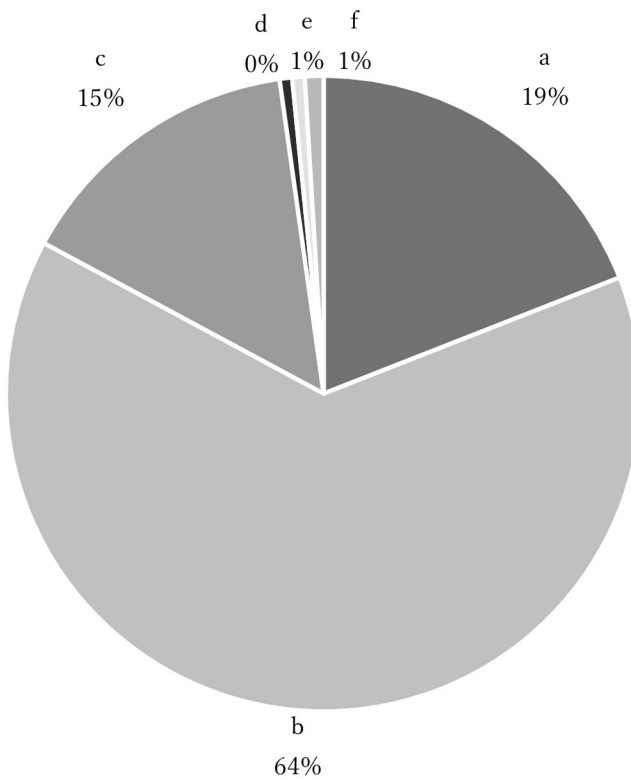

a. In doubt about the effectiveness of the vaccine
b. Being afraid of side response/in doubt about safety
c. Being resistant to the vaccine in general
d. Having history of anaphylactic shock
e. Having underlying illness
f. Other ("Vaccine is not necessary to fight against the COVID-19.")

**Fig 2. Reason of having no intention to vaccinate themselves (N = 315).**

**Table 3. Crude and adjusted odds ratios of having no intention to vaccine their child (N = 1,200).**

| Variable | Crude OR (95% CI) | Adjusted OR[1] (95% CI) |
|---|---|---|
| **Trusted source of information** | | |
| Government/public organization | 1.00 | 1.00 |
| Public news media (NHK) | 0.97 (0.64–1.45) | 1.06 (0.69–1.63) |
| Private news media | 1.45 (1.02–2.07) | 1.49 (1.01–2.19) |
| Social media | 3.32 (1.88–5.84) | 2.80 (1.53–5.12) |
| Word of mouths | 2.63 (1.65–4.18) | 2.41 (1.46–3.95) |
| No source | 2.77 (1.95–3.94) | 2.87 (1.94–4.26) |
| **Parent's gender** | | |
| Male | 1.00 | 1.00 |
| Female | 2.51 (1.97–3.21) | 2.43 (1.57–3.74) |
| **Parent's age (years)** | | |
| < 34 | 1.00 | 1.00 |
| 35–39 | 0.63 (0.41–0.99) | 1.61 (0.96–2.71) |
| 40–45 | 0.52 (0.34–0.81) | 1.11 (0.76–1.62) |
| ≥ 45 | 0.35 (0.23–0.53) | 1.06 (0.76–1.49) |
| **Annual household income in 2020 (million yen)** | | |
| <4 | 1.43 (1.01–2.03) | 1.09 (0.69–1.70) |
| 4–6 | 1.35 (1.02–1.79) | 1.19 (0.86–1.66) |
| ≧6 | 1.00 | 1.00 |
| Unknown | 2.72 (1.64–4.49) | 1.91 (1.09–3.34) |
| **Parent's job type** | | |
| Employed/self-employed | 1.00 | 1.00 |
| Part-time | 1.23 (0.87–1.77) | 0.63 (0.40–1.00) |
| Unemployed/homemaker/student | 2.19 (1.66–2.90) | 1.08 (0.73–1.61) |
| **Partner's job type** | | |
| Employed/self-employed | 1.00 | 1.00 |
| Part-time | 0.51 (0.36–0.72) | 0.95 (0.61–1.46) |
| Unemployed/homemaker/student | 0.42 (0.29–0.60) | 0.68 (0.43–1.07) |
| No partner | 1.11 (0.61–2.03) | 0.90 (0.44–1.83) |
| **Parent's work place** | | |
| Work at home everyday | 1.00 | -[2] |
| Work at home half of a week | 0.69 (0.38–1.24) | |
| Work away from home | 1.23 (0.81–1.88) | |
| Unemployed/homemaker/student | 2.31 (1.47–3.62) | |
| **Partner's work place** | | |
| Work at home everyday | 1.00 | -[2] |
| Work at home half of a week | 1.08 (0.60–1.95) | |
| Work away from home | 1.15 (0.74–1.78) | |
| Unemployed/homemaker/student | 0.53 (0.31–0.90) | |
| No partner | 1.43 (0.70–2.92) | |
| **Fired or salary reduction after the outbreak (parent)** | | |
| No | 1.00 | 1.00 |
| Yes | 0.97 (0.74–1.28) | 1.22 (0.86–1.73) |
| **Fired or salary reduction after the outbreak (partner)** | | |
| No | 1.00 | 1.00 |
| Yes | 0.94 (0.68–1.30) | 0.66 (0.44–1.00) |
| **Parent's satisfaction to the social relationship** | | |

*(Continued)*

**Table 3.** (Continued)

| Variable | Crude OR (95% CI) | Adjusted OR[1] (95% CI) |
|---|---|---|
| High | 1.00 | 1.00 |
| Medium | 1.02 (0.74–1.40) | 1.04 (0.73–1.47) |
| Low | 1.05 (0.73–1.50) | 0.96 (0.63–1.45) |
| **Perceived risk of infection** | | |
| High | 1.00 | 1.00 |
| Medium | 0.87 (0.66–1.14) | 1.02 (0.73–1.42) |
| Low | 1.47 (1.07–2.02) | 1.55 (1.04–2.32) |
| **Distrust of the health care system** | | |
| High | 1.00 | 1.00 |
| Medium | 1.27 (0.87–1.86) | 1.15 (0.73–1.42) |
| Low | 1.14 (0.84–1.55) | 1.03 (0.69–1.51) |
| **Parent's mental distress (K6 score)** | | |
| None (0–4) | 1.00 | 1.00 |
| Moderate (5–9) | 0.99 (0.73–1.36) | 0.96 (0.68–1.36) |
| Severe ($\geq$ 10) | 1.16 (0.88–1.53) | 1.05 (0.74–1.47) |
| **Concerns about household economic situation** | | |
| High | 1.00 | 1.00 |
| Medium | 0.91 (0.62–1.33) | 0.81 (0.52–1.25) |
| Low | 0.90 (0.64–1.27) | 0.81 (0.52–1.26) |
| **Concerns about own and family member's stress** | | |
| High | 1.00 | 1.00 |
| Medium | 1.34 (0.90–1.99) | 1.29 (0.82–2.04) |
| Low | 1.26 (0.88–1.79) | 1.23 (0.79–1.93) |
| **Concerns about lifestyle of their child** | | |
| High | 1.00 | 1.00 |
| Medium | 0.67 (0.43–1.05) | 0.62 (0.37–1.02) |
| Low | 0.95 (0.66–1.35) | 0.90 (0.58–1.40) |
| **Size of city of residence (population)** | | |
| < 50,000 | 1.00 | 1.00 |
| 50,000–100,000 | 1.17 (0.81–1.71) | 1.13 (0.74–1.72) |
| 100,000–300,000 | 1.10 (0.81–1.49) | 1.12 (0.78–1.59) |
| 300,000–500,000 | 1.00 (0.68–1.48) | 0.95 (0.61–1.48) |
| $\geq$ 500,000 | 0.99 (0.67–1.47) | 0.78 (0.49–1.25) |
| **State/Semi-state of emergency is declared at the time of survey** | | |
| Yes | 1.00 | 1.00 |
| No | 0.93 (0.71–1.22) | 0.90 (0.66–1.25) |
| **Living with parents** | | |
| No | 1.00 | 1.00 |
| Yes | 1.13 (0.74–1.73) | 1.42 (0.88–2.28) |
| **Number of children living together** | | |
| 1 | 1.00 | 1.00 |
| 2 | 1.16 (0.89–1.51) | 1.15 (0.86–1.55) |
| $\geq$ 3 | 0.93 (0.64–1.35) | 0.88 (0.58–1.34) |
| **Child's gender** | | |
| Male | 1.00 | 1.00 |
| Female | 1.27 (1.00–1.61) | 1.26 (0.97–1.63) |
| **Child's age (years)** | | |

(*Continued*)

**Table 3.** (Continued)

| Variable | Crude OR (95% CI) | Adjusted OR[1] (95% CI) |
|---|---|---|
| 3–5 | 1.00 | 1.00 |
| 6–11 | 0.64 (0.49–0.85) | 0.72 (0.52–0.99) |
| ≥ 12 | 0.48 (0.35–0.67) | 0.57 (0.38–0.85) |

OR: Odds ratio, CI: Confidence interval.

[1]Adjusted for all other variables in the table.

[2]Omitted due to multicollinearity.

on Twitter tended to believe conspiracy theories and engage in the community [24]. Although media specific behavior have been reported [14,23], the present study was not able to identify which social media was associate with parental vaccine hesitancy.

Several studies reported women were less likely to be willing to vaccinate their child [25–27]. Also women were more likely to hesitate to vaccinate themselves globally [15,28]. Although the present study did not have data to assess reasons of the gender difference, it may be due to fear of adverse reaction. Women have more chance to experience adverse reaction, and there are unreliable news about the relationship between vaccination and infertility [29]. Interestingly, the results showed that the gender difference related to vaccine hesitancy attenuated when mothers had high satisfaction regarding social relationship. It has been reported that social-distancing and increased anxiety due to the pandemic has increased internet use globally [30,31]. Increased use of smartphones was associated with increased game use among men while increased use of social networking services was observed among women [32,33]. This may explain why women who had limited social relationships tended to spend more time using social media and obtained vaccine-related information via social network. The reasons leading to this significant gender difference need to be further investigated, and gender-specific communication strategies to promote vaccination among children should be considered.

The strength of this study is that the study covered the whole nation and focused on parents with children. While studies on parental vaccine hesitancy is limited in Japan, the COVID-19 vaccine program was recently expanded to children >12 years of age. The findings will be useful to promote vaccination in the country. However, the present study is not without limitations. Owing to the nature of the data collection via an internet-based survey and participation is on voluntary basis, this study is subject to selection bias. Generalizability may be limited, as those who participated may be more interested in the COVID-19 vaccine and have different attitude toward the vaccination. Also, those who had access to internet might be different from those who did not. Therefore, the percentage of vaccine hesitancy may not be representative of the general population, however, identified factors related to vaccine hesitancy can be used to

**Table 4. Adjusted odds ratios of having no intention to vaccinate their child in relation to parent's satisfaction to the social relationship by their gender (N = 1,200).**

| Variable | Adjusted OR (95% CI) | | p-value (LRT) |
|---|---|---|---|
| | Parent's gender | | |
| | Male | Female | |
| Parent's satisfaction to the social relationship | | | |
| High | 1.00 | 1.29 (0.65–2.55) | 0.021 |
| Low/Medium | 0.66 (0.40–1.07) | 1.86 (1.03–3.37) | |

OR: Odds ratio, CI: Confidence interval, LRT: Likelihood ratio test.

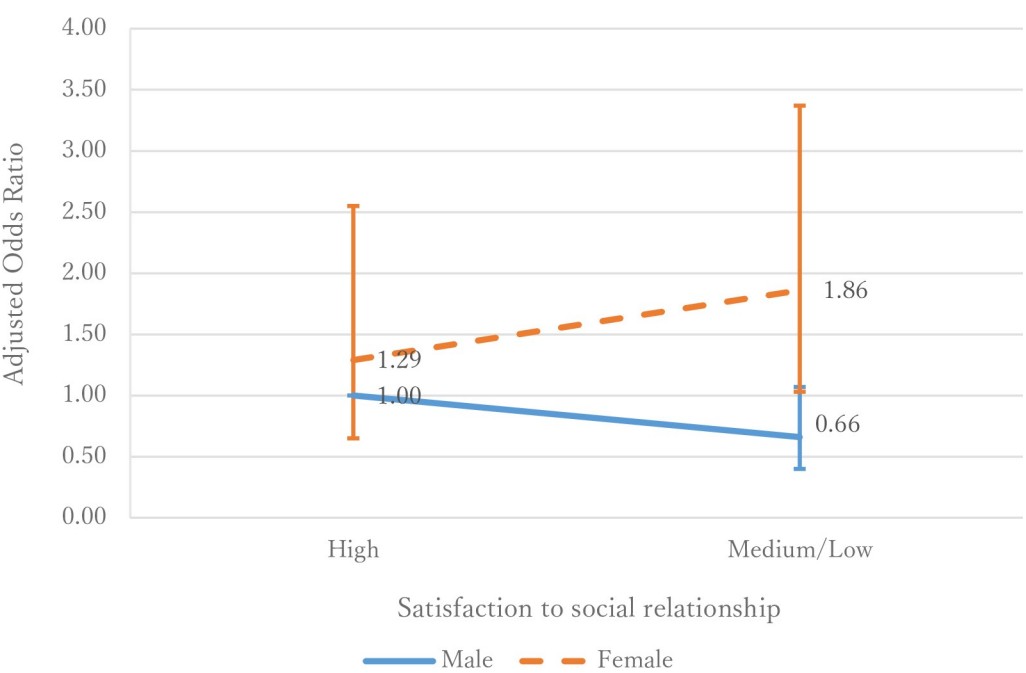

**Fig 3. Adjusted odds ratios of having no intention to vaccinate their child in relation to parent's satisfaction to the social relationship by their gender in Japan (year 2021).**

consider targeting people who have active access to internet. Secondly, this study did not ask details about which social media was mainly used though some studies reported that effect on vaccine hesitancy varied by different types of social media such as Youtube, Facebook and Twitter [14,23]. Regardless of the limitations, the results were consistent with findings in other countries, which support reliability of the findings.

Although children have been considered to develop mild symptoms compared to adults, recent studies have reported that COVID-19-related multisystem inflammatory syndrome can cause morbidity and mortality [34,35]. The recent rapid increase of cases among children could lead to an increase of severe cases in this age group. Rapid expansion of vaccine coverage including children is important to protect individuals as well as communities.

To address parental vaccine hesitancy, it is necessary to consider how to disseminate correct information other than through public channels to reach people who seek information from social media. Government officials should consider new channels such as social media including collaboration with influencers to disseminate information other than traditional methods such as posting message on their homepages. We also found that mothers are more likely to hesitate to vaccinate their children than fathers, especially among people who had lower satisfaction to social relationship. Therefore, healthcare professionals and government officials need to carefully consider this gender difference, and provide targeted information especially for those who are likely to be socially isolated due to measures of social distancing.

## Conclusion

The results showed trust in social media as source of information, being a mother, and lower perceived risk of infection was related to higher risk of parental vaccine hesitancy. The gender difference attenuated among parents with high satisfaction toward social relationships. It is necessary to consider how to disseminate correct information other than through public

channels to reach people who seek information from social media. Also, healthcare profession-
als and local government officials need to carefully consider the gender difference and provide
targeted information.

## Supporting information

**S1 File. Questionnaire, English translation.**
(DOCX)

**S2 File. Minimal data set.**
(CSV)

## Acknowledgments

The data collection was supported by the Nippon Research Center.

## Author Contributions

**Conceptualization:** Sayaka Horiuchi, Haruka Sakamoto, Ryoji Shinohara.

**Formal analysis:** Sayaka Horiuchi.

**Supervision:** Zentaro Yamagata.

**Visualization:** Sayaka Horiuchi.

**Writing – original draft:** Sayaka Horiuchi.

**Writing – review & editing:** Haruka Sakamoto, Sarah K. Abe, Ryoji Shinohara, Megumi
    Kushima, Sanae Otawa, Hideki Yui, Yuka Akiyama, Tadao Ooka, Reiji Kojima, Hiroshi
    Yokomichi, Kunio Miyake, Takashi Mizutani, Zentaro Yamagata.

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
