## [Decision Letter · Decision Letter 0]

15 Nov 2021

PONE-D-21-34773Factors of parental COVID-19 vaccine hesitancy: a cross sectional study in JapanPLOS ONE

Dear Dr. Horiuchi,

Thank you for submitting your manuscript to PLOS ONE. After careful consideration, we feel that it has merit but does not fully meet PLOS ONE’s publication criteria as it currently stands. Therefore, we invite you to submit a revised version of the manuscript that addresses the points raised during the review process. Please submit your revised manuscript by 20 November 2021. If you will need more time than this to complete your revisions, please reply to this message or contact the journal office at plosone@plos.org. Please include the following items when submitting your revised manuscript:A rebuttal letter that responds to each point raised by the academic editor and reviewer(s). You should upload this letter as a separate file labeled 'Response to Reviewers'.A marked-up copy of your manuscript that highlights changes made to the original version. You should upload this as a separate file labeled 'Revised Manuscript with Track Changes'.An unmarked version of your revised paper without tracked changes. You should upload this as a separate file labeled 'Manuscript'.

We look forward to receiving your revised manuscript.

Kind regards,

Bijaya Kumar Padhi, PhD, MPH

Academic Editor

PLOS ONE

Additional Editor Comments:

This manuscript investigates parental COVID-19 vaccine hesitancy and its associated factors in Japan. The study has a valid research question and hypothesis. Language and and presentation are clear and adequate, figures and tables are in line with scientific norms and standards. I commend the authors for conceptualizing this study, as the findings certainly add value to the literature. I have following comments:

Psychometrics of the survey instruments should be provided.Authors said they used multiple logistic regression model. However, there is a lack of clarity on what basis the model was built. Whether they used a-priori hypothesis or any stepwise input of parameters. This information should be provided in data analysis section.

Reviewers' comments:

Reviewer's Responses to Questions

**Comments to the Author**

1. Is the manuscript technically sound, and do the data support the conclusions?

Reviewer #1: Yes

Reviewer #2: Yes

2. Has the statistical analysis been performed appropriately and rigorously? 

Reviewer #1: Yes

Reviewer #2: Yes

3. Have the authors made all data underlying the findings in their manuscript fully available?

Reviewer #1: Yes

Reviewer #2: Yes

4. Is the manuscript presented in an intelligible fashion and written in standard English?

Reviewer #1: Yes

Reviewer #2: Yes

5. Review Comments to the Author

Reviewer #1: This subject is absolutely essential for disease prevention and socio-behavioral health.

Introduction

Well written

Methods

The participant section provides the detailed registration system of the Nippon Research Center. What is the objective of the registration? Which types of communities are registering? Are both parents reporting? Till the study, how many registered? As it is voluntary, registration may be for those who have positive motivation, they registered. How did you justify your representation of sampling? If possible, clarify, otherwise mention this in limitation.

In the data collection section: 1200 is a good sample. How did you select this sample from the register? What is your target population?

Results

In line 180

Intention to vaccinate themselves (parent), since adult vaccination started long back, it is good to presented vaccinated versus non-vaccinated (why intention to vaccinate) – present parent practice rather than their attitude. Then you can associate parent vaccination status with intension to vaccinate their child.

Table 3. Reason of having no intention to vaccinate themselves (N = 315) – present the finding using a figure (pi chart).

Combine Table 1 and Table 4 (The variable which described in one table describe it in text)

Table 1. Characteristics of study parents in Japan (year 2021) (N = 1,200)

Table 4. Crude and adjusted odds ratios of having no intention to vaccine their child 202 (N = 1,200)

Discussion

Add one paragraph on Implication for policy and practice

Reviewer #2: the manuscript s well written with the methodology, results section in aligment to the objective. the discussion part is throughly written and well explained. one suggestion in the analysis section is while looking at level of social relationship satisfaction related to parental vaccine hesitancy, high satifaction can be kept as reference

6. PLOS authors have the option to publish the peer review history of their article (what does this mean?). If published, this will include your full peer review and any attached files.

Reviewer #1: **Yes: **Dr. Krushna Chandra Sahoo, ICMR-Regional Medical Research Centre, Bhubaneswar, India

Reviewer #2: No

---

## [Author Response · Author response to Decision Letter 0]

19 Nov 2021

Dear Dr Bijaya Kumar Padhi:

Thank you for giving me the opportunity to submit a revised draft of my manuscript to PLOS ONE titled “Factors of parental COVID-19 vaccine hesitancy: a cross sectional study in Japan.” We appreciate the time and effort that you and the reviewers have dedicated to providing your valuable feedback on my manuscript. We are grateful to the reviewers for their insightful comments on our paper. We have been able to incorporate changes to reflect most of the suggestions provided by the reviewers and have indicated the changes within the manuscript in red font. Here is a point-by-point response to the reviewers’ comments and concerns.

Editor:

Reply: We have carefully reviewed the guidelines and revised the manuscript accordingly.

Reply: We confirm that minors were not included in the study participants. We have added explanation on online informed consent as follows.

Lines 84-85 “Participants clicked the checkbox on the screen online to provide informed consent.”

Additional Editor Comments:

This manuscript investigates parental COVID-19 vaccine hesitancy and its associated factors in Japan. The study has a valid research question and hypothesis. Language and and presentation are clear and adequate, figures and tables are in line with scientific norms and standards. I commend the authors for conceptualizing this study, as the findings certainly add value to the literature. I have following comments:

1. Psychometrics of the survey instruments should be provided.

2. Authors said they used multiple logistic regression model. However, there is a lack of clarity on what basis the model was built. Whether they used a-priori hypothesis or any stepwise input of parameters. This information should be provided in data analysis section.

Reply: We appreciate your feedback. 

1. We have added the survey instrument as a supporting information file. 

2. We selected covariates based on a-priori hypotheses, and attempted to include all of them variables in the multivariate analysis. Finally, two variables were excluded from the model due to multicolinearity. We have added explanation as follows.

Lines 157-159 “We selected variables to be included in the multivariate model based on a-priori hypothesis, and included all covariates in the model unless there was an evidence of multicollinearity.”

Reviewer 1:

This subject is absolutely essential for disease prevention and socio-behavioral health.

Introduction

Well written

Methods

The participant section provides the detailed registration system of the Nippon Research Center. What is the objective of the registration? Which types of communities are registering? Are both parents reporting? 

Reply: The Nippon Research Center is a private entity that subcontracts public and private surveys. To respond needs of its customers, it has built several panels and recruited participants from whole nation. This time, we used the panel for web-based surveys. 

Only one parent answered the survey. To avoid confusion, we have rephrased parents into other words in the participants section as follows. 

Lines 95-97 “The eligibility criteria were 1) those who had children aged 3–14 years, and 2) those who resided in Japan at the time of inception of the survey. We did not set an age limit for participants.”

Lines 101-104 “The Nippon Research Center is a private entity that subcontracts public and private surveys. To respond needs of its customers, it has built several panels and recruited participants from whole nation.”

Till the study, how many registered? 

Reply: Information of the panel has been added as follows.

Line 104 “As of October 2020, about 1.4 million registered in the panel.”

As it is voluntary, registration may be for those who have positive motivation, they registered. How did you justify your representation of sampling? If possible, clarify, otherwise mention this in limitation.

Reply: We agree that the selected participants were not representative of the Japanese population due to nature of the internet survey. We have included this point as limitation.

Lines 263-270 “Owing to the nature of the data collection via an internet-based survey and participation is on voluntary basis, this study is subject to selection bias. Generalizability may be limited, as those who participated may be more interested in the COVID-19 vaccine and have different attitude toward the vaccination. Also, those who had access to internet might be different from those who did not. Therefore, the percentage of vaccine hesitancy may not be representative of the general population, however, identified factors related to vaccine hesitancy can be used to consider targeting people who have active access to internet.”

In the data collection section: 1200 is a good sample. How did you select this sample from the register? What is your target population?

Reply: Participation to this survey was on voluntary basis. We asked people registered in the panel to join this survey and continued recruitment until the number reached 1200. 

Lines 112-113 “People registered in the panel participated in this survey on a voluntary basis in response to an invitation. Data collection continued until the sample size reached 1,200.”

Results

In line 180

Intention to vaccinate themselves (parent), since adult vaccination started long back, it is good to presented vaccinated versus non-vaccinated (why intention to vaccinate) – present parent practice rather than their attitude. Then you can associate parent vaccination status with intension to vaccinate their child.

Table 3. Reason of having no intention to vaccinate themselves (N = 315) – present the finding using a figure (pi chart).

Reply: We appreciated your suggestion. As we did not have data on vaccine status, it is impossible to analyze data based on parental vaccination status. The eligibility of the COVID-19 vaccine was expanded to people under 65 years of age in May 2021. As this survey was conducted May 25 and June 3, 2021, we decided to ask intention to vaccine assuming that the proportion of vaccinated was very low at the time of survey.

We have added “Fig2 Reason of having no intention to vaccinate themselves (N = 315)” according to your suggestion.

Combine Table 1 and Table 4 (The variable which described in one table describe it in text)

Table 1. Characteristics of study parents in Japan (year 2021) (N = 1,200)

Table 4. Crude and adjusted odds ratios of having no intention to vaccine their child 202 (N = 1,200)

Reply: We appreciate your suggestion, however, we are afraid that, if combined, Table will get very large and difficult to see for readers. We, therefore, would like to keep separate tables for reporting characteristics of participants and odds ratios. We hope it will be acceptable.

Discussion

Add one paragraph on Implication for policy and practice

Reply: We have added the following sentences in the last paragraph of the discussion section.

Lines 281-290 “To address parental vaccine hesitancy, it is necessary to consider how to disseminate correct information other than through public channels to reach people who seek information from social media. Government officials should consider new channels such as social media including collaboration with influencers to disseminate information other than traditional methods such as posting message on their homepages. We also found that mothers are more likely to hesitate to vaccinate their children than fathers, especially among people who had lower satisfaction to social relationship. Therefore, healthcare professionals and government officials need to carefully consider this gender difference, and provide targeted information especially for those who are likely to be socially isolated due to measures of social distancing.”

Reviewer 2: 

the manuscript s well written with the methodology, results section in aligment to the objective. the discussion part is throughly written and well explained. one suggestion in the analysis section is while looking at level of social relationship satisfaction related to parental vaccine hesitancy, high satifaction can be kept as reference

Reply: We appreciate your insightful suggestion. To make a reference group as the group at the lowest risk of vaccine hesitancy like other variables in the model, we set the high satisfaction group as reference for univariate, multivariate logistic models as well as interaction analyses. In the interaction analysis, we combined low and middle satisfaction groups to make the statistical analyses stable. The revised results are shown in table 4 and figure 3.

Lines 35-38 “People with lower satisfaction to social relationships tended to be more hesitant to vaccinate their child among mothers in contrast to fathers who showed constant or lower hesitancy to vaccinate their child in the lower satisfaction group (LRT p=0.021).”

Lines 216-219 “While fathers had constant or more intention to vaccinate their child at the lower level of satisfaction, mothers were more likely to hesitate to vaccinate their child when they had lower satisfaction regarding social relationship (Figure 3).”

---

## [Editor Report · Decision Letter 1]

29 Nov 2021

Factors of parental COVID-19 vaccine hesitancy: a cross sectional study in Japan

PONE-D-21-34773R1

Dear Dr. Horiuchi,

We’re pleased to inform you that your manuscript has been judged scientifically suitable for publication and will be formally accepted for publication once it meets all outstanding technical requirements.

Kind regards,

Bijaya Kumar Padhi, PhD, MPH

Academic Editor

PLOS ONE

Additional Editor Comments (optional):

Thank you for submitting the revised version of your manuscript to PLOS ONE. After careful consideration, we feel that it has merit to meet PLOS ONE’s publication criteria.
---

## [Editor Report · Acceptance letter]

9 Dec 2021

PONE-D-21-34773R1 

Factors of parental COVID-19 vaccine hesitancy: a cross sectional study in Japan 

Dear Dr. Horiuchi:

I'm pleased to inform you that your manuscript has been deemed suitable for publication in PLOS ONE. Congratulations! Your manuscript is now with our production department. 

Kind regards, 

on behalf of

Dr. Bijaya Kumar Padhi 

Academic Editor

PLOS ONE